# Alcohol potentiates a pheromone signal in flies

Annie Park[1†]*, Tracy Tran[1], Elizabeth A Scheuermann[2], Dean P Smith[2], Nigel S Atkinson[1]*

[1]Department of Neuroscience and Waggoner Center for Alcohol and Addiction Research, The University of Texas at Austin, Austin, United States; [2]Department of Pharmacology and Neuroscience, University of Texas Southwestern Medical Center, Dallas, United States

**Abstract** For decades, numerous researchers have documented the presence of the fruit fly or *Drosophila melanogaster* on alcohol-containing food sources. Although fruit flies are a common laboratory model organism of choice, there is relatively little understood about the ethological relationship between flies and ethanol. In this study, we find that when male flies inhabit ethanol-containing food substrates they become more aggressive. We identify a possible mechanism for this behavior. The odor of ethanol potentiates the activity of sensory neurons in response to an aggression-promoting pheromone. Finally, we observed that the odor of ethanol also promotes attraction to a food-related citrus odor. Understanding how flies interact with the complex natural environment they inhabit can provide valuable insight into how different natural stimuli are integrated to promote fundamental behaviors.

*For correspondence:
annie.park@dpag.ox.ac.uk (AP);
nigela@utexas.edu (NSA)

Present address: †Centre for Neural Circuits and Behaviour, The University of Oxford, Oxford, United Kingdom

Competing interests: The authors declare that no competing interests exist.

## Introduction

Conflict that results in aggression occurs across the animal kingdom. Aggression in *Drosophila melanogaster* is a well-documented behavior; studies have identified several aggression-regulating pheromones, circuits, and genes (*Wang and Anderson, 2010*; *Vrontou et al., 2006*; *Nilsen et al., 2004*; *Asahina et al., 2014*; *Dow and von Schilcher, 1975*). The most well-studied pheromone, 11-cis-Vaccenyl acetate (cVa) is produced by males and has been shown to increase aggression in male flies (*Wang and Anderson, 2010*). Most studies of cVa use this pheromone in isolation by adding it to a behavioral arena or painting flies with it. However, in the wild, flies will encounter cVa when they aggregate on fermenting fruits where they will experience cVa in combination with volatile compounds produced by fermentation (*Zhu et al., 2003*; *Keesey et al., 2016*). Despite the ecological complexity of the fruit fly niche, little is understood about how ethologically relevant combinations of odors influence the underlying neurobiology of behavior.

Ethanol is one of the products of fermentation and fruit flies in particular are attracted to alcohol-containing fruits (<7% ethanol by volume) (*McKenzie and McKechnie, 1979*). Remarkably, there is no identified canonical ethanol olfactory receptor despite the fact that a wide range of ethanol-related behaviors have been identified in flies (reviewed in *Park et al., 2017*). *Kim et al., 1998* identified an odorant-binding protein known as LUSH that mediates aversion to high concentrations of alcohol. LUSH was the first protein shown to directly bind ethanol (*Kruse et al., 2003*). In the fly antennae, the only neurons known to respond to cVa are also those that express LUSH (*Xu et al., 2005*). However, these neurons do not show any electrophysiological responses to ethanol alone (*Figure 1a* and *Kim et al., 1998*). We hypothesized that these neurons could respond to the combination of ethanol and cVa. Here, we ask if ethanol potentiates the cVa signal to enhance inter-male aggression.

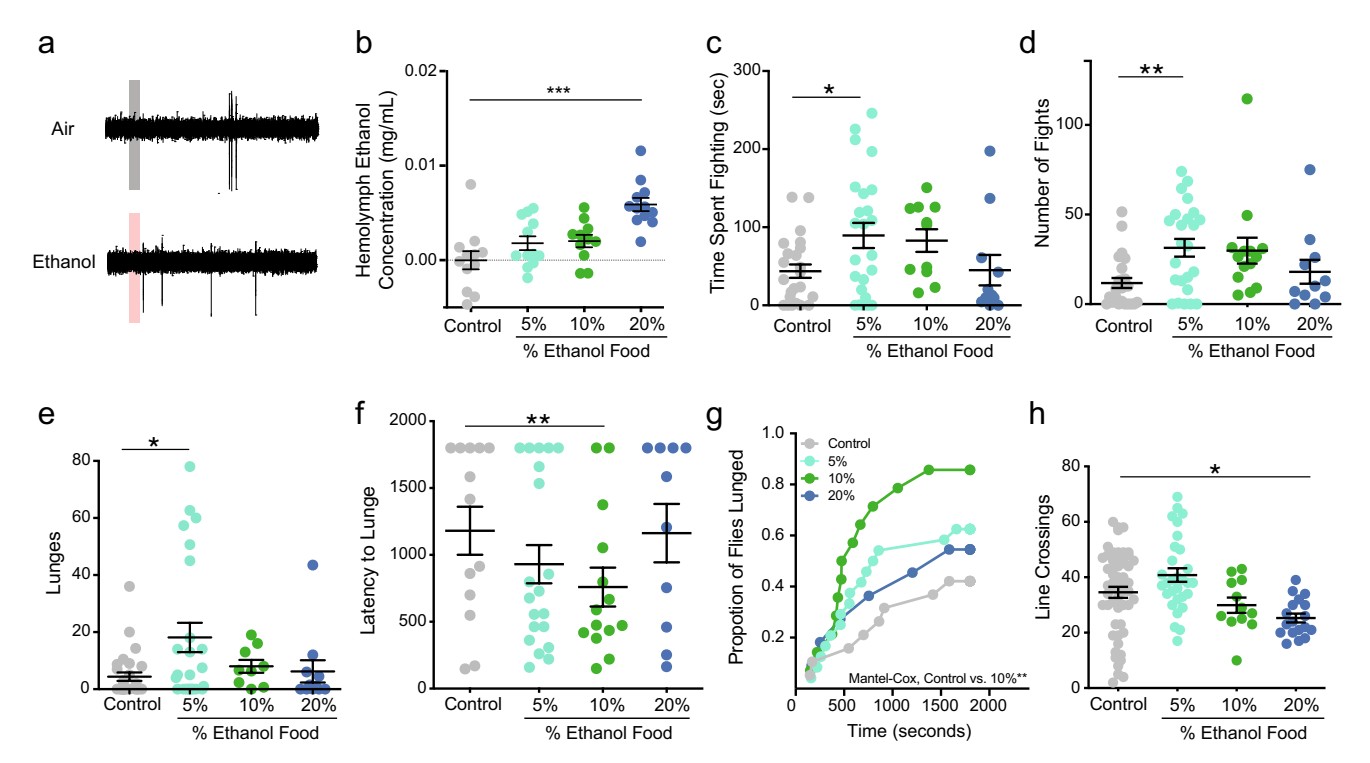

**Figure 1.** Alcohol odor increases aggression in male flies. (**a**) Traces of T1 sensilla recordings with a 300 ms exposure to air or vapor from 30% ethanol. (**b**) Hemolymph ethanol concentration (mg/mL) in flies in aggression arenas for 30 mins show no significant increases except with 20% ethanol (one-way ANOVA with Dunnett's p<0.0001, n = 11–12). (**c**) Time spent fighting on ethanol-containing food (Control vs. 5% p=0.038 Kruskal-Wallis test with Dunn's correction, n = 10–20). (**d**) Number of fights on ethanol-containing food (Control vs. 5% p=0.0012, statistical tests as in c). (**e**) Lunges on ethanol-containing food (Control vs. 5% p=0.0225, statistical tests as in c). (**f**) Latency to lunge (p=0.009, Log-rank Mantel-Cox with Bonferroni correction). (**g**) Cumulative latency of flies that lunged during the test (Control vs. 10% p=0.009, Log-rank Mantel-Cox with Bonferroni correction). (**h**) Locomotion as measured by line crossings during the test (Control vs. 20% p=0.0048, statistical tests as in 1 c). p<0.05 *; p<0.01 **; p<0.001 ***. Error bars denote SEM.

Using fly aggression arenas, we added ethanol to fly food in amounts of 5%, 10%, and 20% by volume (*Mundiyanapurath et al., 2006*). To determine the fly's level of aggression we video recorded two wild-type (Canton-S) males in the arena for 30 min and scored fighting events manually. We catalogued fencing, shoving, boxing, tussling, and lunging as aggressive behaviors (described in *Chen et al., 2002* as offensive actions). We measured hemolymph or 'blood' ethanol concentration (BEC), after the flies were in the arena for 30 min (*Figure 1b*), to determine if the flies were receiving an intoxicating dose of ethanol. Neither the 5% nor the 10% ethanol-containing foods caused any detectable rise in ethanol in the flies.

Males that fought on 5% ethanol-containing food exhibited an increase in time spent fighting, number of fights they engaged in, and number of lunges executed (*Figure 1c,d and e*). Although, males on 10% ethanol food exhibited an increase in the proportion that lunged and decreased latency to lunge, they did not show an increase in total number of lunges executed compared to control animals (*Figure 1e,f & g*). These data demonstrate that when flies occupy food patches with ethologically relevant concentrations of ethanol, they display elevated levels of aggression that are not due to increased locomotion (*Figure 1h*). One possible explanation for the reduction of aggression in flies that fought on 20% ethanol compared to those on 5% ethanol can be explained by *Wang and Anderson, 2010* observation that males respond to cVa in a dose-dependent manner with low concentrations of cVa promoting aggregation and aggression, whereas high concentrations of cVa cause dispersion (*Wang and Anderson, 2010*). Twenty-percent ethanol may have potentiated the cVa signal so that it mimics a high cVa concentration causing males to disperse and spend less time fighting.

We sought to determine if ethanol influences the neuronal responses to cVa pheromone by monitoring activity of the T1 cVa sensing neurons in the presence of mixtures of these two odorants. We recorded from cVa-responsive neurons while exposing them to 1% cVa and 5% or 30% ethanol (cVA concentration chosen for comparison to prior literature; ethanol concentrations chosen because they were commonly used in other experiments; *Figure 2a*). Although ethanol does not acutely activate these neurons, we found that it substantially enhanced their cVa responses (*Figure 2b*). We found that 30% ethanol substantially increased cVa-evoked activity (Δ Spikes), while 5% ethanol increased activity to a lesser extent (*Figure 2c,e,f and g*). One possible explanation for why we did not observe an increase in Δ Spikes with cVa and 5% ethanol even though 5% ethanol increased aggression is because the duration of exposure was considerably shorter (1 min) compared to the behavioral assay (30 min), which may not have allowed for sufficient accumulation of alcohol. These T1 neurons have an unstimulated spontaneous firing rate of approximately 1 Hz and increase their firing in response to cVa (*Xu et al., 2005*). Spontaneous responses consistently increased when we applied 5% or 30% ethanol (*Figure 2d*). Finally, the time constant to deactivation increased with 30% ethanol application indicating that neurons had greater sustained activity following cVa treatment (*Figure 2h*). These data are consistent with the notion that ethanol increases inter-male aggression by potentiating responses to cVa.

In the wild, ethanol is almost always present with other fruit volatiles and fermentation odorants. Farnesol is an odorant present in the rinds of citrus fruits, which are known to be attractive to flies (*Ronderos et al., 2014*; *Dweck et al., 2013*). We asked if ethanol would potentiate the electrophysiological response to farnesol and attraction to farnesol. We chose farnesol because Or83c neurons are not broadly tuned and display strong activation to farnesol (similar to how cVa-sensing neurons are only tuned to cVa) and do not show a response to ethanol alone (*Ronderos et al., 2014*). Farnesol also conveys an entirely different behavioral response from cVa. We performed single-sensillum recordings (SSR) of the ai2 sensilla while exposing them to 30% ethanol and recorded their responses to farnesol (*Figure 3a*). We found that both evoked activity and spontaneous activity

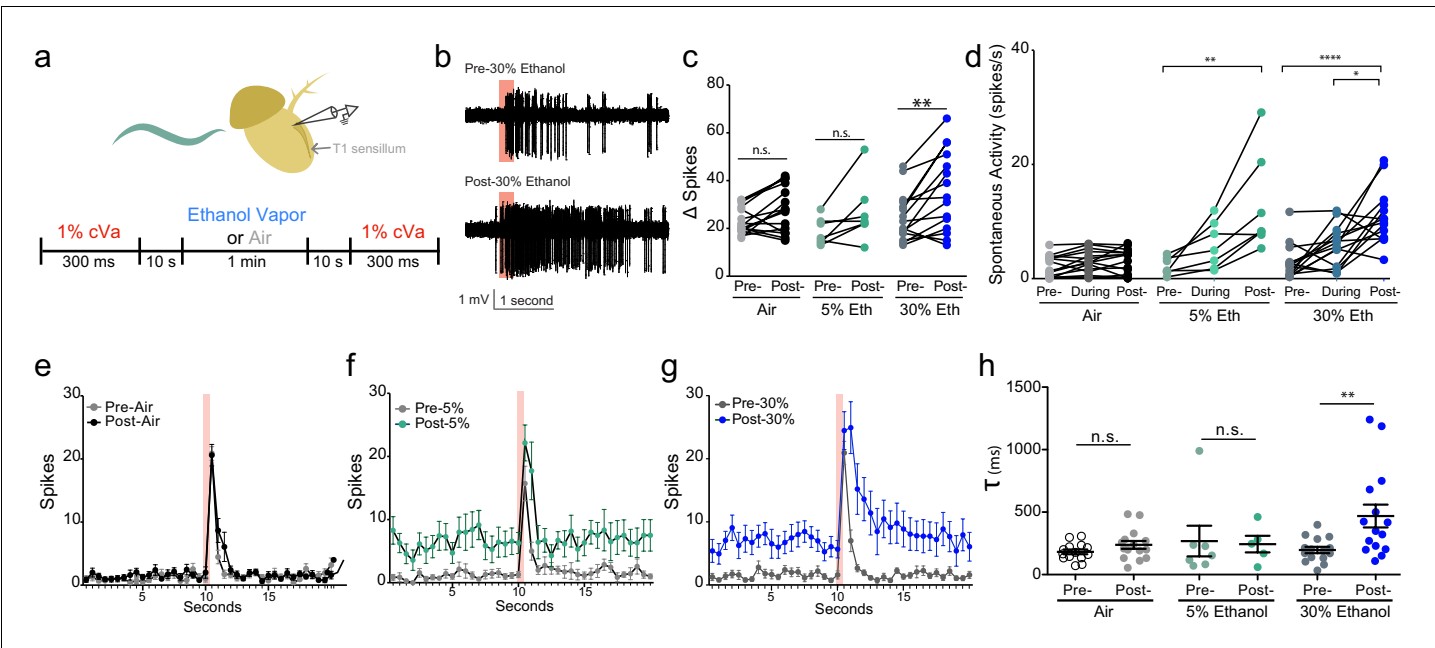

**Figure 2.** Alcohol odor potentiates the response to cVa. (**a**) Experimental timeline and diagram of recording site on fly antenna. (**b**) Traces of cVa-sensing T1 neurons. The red bar denotes 300 ms cVa exposure. (**c**) Δ Spikes calculated as cVa-induced activity (1 s during and after cVa) – spontaneous activity (paired two-tailed t-test, p=0.002, n = 7,15,15). (**d**) Spontaneous activity before, during, and after ethanol exposure. Spontaneous activity calculated as the total number of spikes 10 s prior to cVa delivery/10 s (Pre- vs. Post-5% p=0.002, Pre- vs. Post-30% p<0.0001, During vs. Post-30% p=0.0152, Kruskal-Wallis test with Dunn's correction). (**e**), (**f**), (**g**) Averaged spikes over time for air, 5% ethanol, and 30% ethanol, respectively. Red bar denotes 300 ms cVa exposure (**h**) Time constant (τ) of decay of the cVa-induced spikes (Mann-Whitney test, p=0.0043). p>0.05, n.s. (not significant); p<0.05 *; p<0.01 **; p<0.0001 ****. Error bars denote SEM.

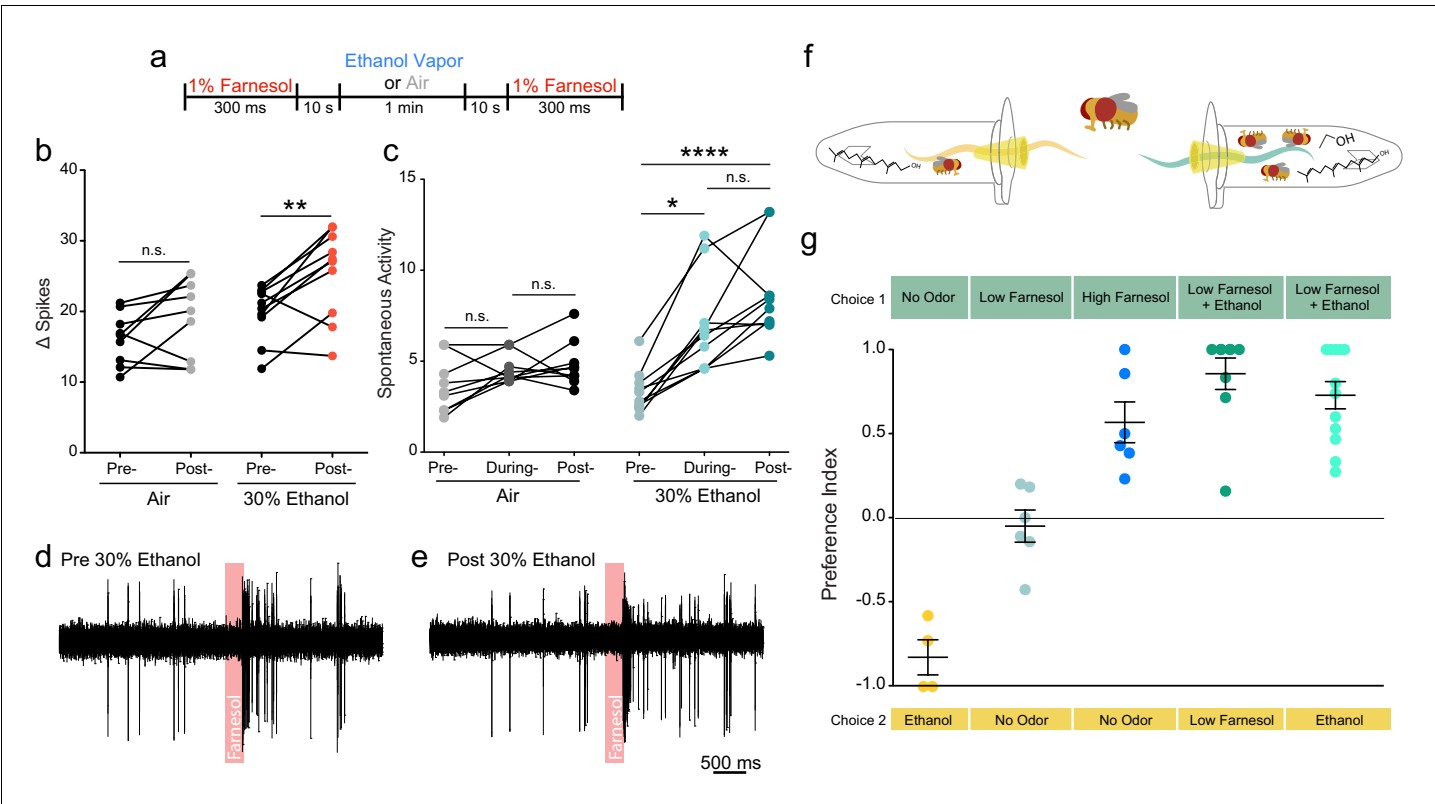

**Figure 3.** Ethanol increases attraction to and potentiates the neuronal response of a food related odor. (a) Paradigm used to evaluate farnesol responses pre- and post-ethanol treatment. Responses are shown in panels b-e. (b) Δ Spikes of farnesol-induced activity in ai2 sensilla, calculated as in 2b (paired two-tailed t-test, p=0.0059, n = 9–10). (c) Spontaneous activity before, during, and after ethanol exposure, calculated as in 2 c (Pre- vs. During Ethanol p=0.017, Pre- vs. Post-Ethanol p<0.0001, Kruskal-Wallis test with Dunn's correction). (d, e) Traces from ai2 farnesol-sensing neurons. The red bar denotes a 300 ms farnesol exposure. (f) Graphic of the two-choice olfactory trap assay used to measure relative attraction to odors. (g) Preference Indices calculated as Number of flies in Choice 1-Number of flies in Choice 2/Total Number of flies. p>0.05, n.s. (not significant); p<0.05 *; p<0.01 **; p<0.0001 ****. Error bars denote SEM.

increased following ethanol treatment (*Figure 3b-e*). To determine if ethanol can augment attraction to farnesol we used two-choice olfactory trap assay (pictured *Figure 3f*). We used a dilution of farnesol that elicited no attraction when used alone ($10^{-5}$). However, when combined with 30% ethanol the odor from a $10^{-5}$ dilution of farnesol displayed much greater attractiveness than either the odor of farnesol or ethanol alone (*Figure 3g*). Flies preferred the mixed farnesol and ethanol odor over farnesol alone or ethanol alone suggesting ethanol potentiates attraction to farnesol.

Previous studies on behavioral responses of *Drosophila* to ethanol have focused on the systemic effects of ethanol. These studies demonstrated that flies, like mammals, become hyperactive with low doses of ethanol, sedate at high doses of ethanol, acquire tolerance to ethanol, display a withdrawal response, and seek ethanol despite negative consequence (reviewed in *Park et al., 2017*). However, ethanol responses described in this paper are fundamentally distinct in that they are most likely olfactory (non-systemic) and are ethologically important to the life of a fly. *Fischer et al., 2017* demonstrated that flies are more attracted to natural mixtures of microbial by-products than the individual components of the mixture. We find evidence that ethanol potentiates two fundamentally distinct odorants and behaviors in *Drosophila*. First, ethanol odor increases cVa signaling, which in turn increases aggression. Second, ethanol augments farnesol signaling, resulting in increased attraction to this food odorant. Female flies are thought to prefer to consume ethanol-laden food and lay their eggs in ethanol-containing food because ethanol has caloric value, antimicrobial properties, and provides a protected niche by suppressing competition from other *Drosophila* species that find the ethanol to be toxic (*Kacsoh et al., 2013*; *Park et al., 2018*; *Azanchi et al., 2013*). When female flies accumulate on the food substrate, males eventually follow, and fight one another for a chance

to mate with the female. Interestingly, male flies are known to deposit cVa directly onto food substrates and will spend more time around the marked area (*Keesey et al., 2016*; *Mercier et al., 2018*). Both the cVa pheromone and farnesol odorant are naturally encountered by flies in the wild and evoke behaviors believed to provide selective advantages. The odor of ethanol combined with a food odor could enhance the perceived value of the food as a reproductive resource, while the combination of the scent of ethanol and cVA could increase the male drive to fight for control of this resource. We documented evidence for potentiated responses to two odors, but it seems likely that ethanol could increase attraction to other food odors and perhaps other fly pheromones as well.

## Materials and methods

### Fly handling

All flies were raised on corn meal malt extract food (7.6%) CH Guenther and Son Pioneer Corn Meal (Walmart, Inc), 7.6% Karo syrup (Walmart, Inc), 1.8% Brewer's yeast (SAF, Milwaukee WI), 0.9% Gelidium agar (Mooragar, Inc, Rocklin, CA), 0.1% nipagin (Fisher Scientific, Inc) in 0.5% ethanol, 11.1% #5888 amber malt extract (Austin Homebrew, Austin, Tx) and 0.5% proprionic acid (Fisher Scientific, Inc). Solids are weight/volume and liquids are volume/volume. Flies were housed a 12:12 light:dark cycle. Flies used in aggression and courtship receptivity behavioral assays were all taken from group housed bottles as pupae and individually raised in vials. Flies used in imaging, immunohistochemistry, and qPCR were group housed.

### Behavioral tests

Aggression chambers were assembled based on description by *Mundiyanapurath et al., 2006*. using a fly vial cut one inch high and glued to one petri dish. The top of the chamber has two holes; one large hole is used for loading flies and one other smaller hole is in the center of the top and is used for circulation. Food wells were made by cutting 1.5 mL microfuge tube tops. Fly food was melted and pipetted into the microfuge tube tops. Ethanol was added into the fly food once the food cooled down to roughly 35°C. We added sucrose to the top of each fly food surface and a decapitated virgin female fly. Flies were loaded into the chamber by gentle aspiration and the video camera began recording 5 min after the flies were in the chamber. Aggression tests were conducted between the hours of 9 AM and 4 PM (Lights on 8 AM – 8PM). Flies tested for aggression were between 4 and 6 days old and Canton S.

Line crossing assays were performed in the aggression chambers. Flies were aspirated into the chamber in pairs and we recorded the total number of line crossings within a 5-min time period. We recorded video from the top of the chambers and drew a line bisecting the chamber. We used Canton S males between 4 and 6 days old.

Olfactory Traps were based on the protocol developed by *Woodard et al., 1989* and were constructed using a 1.5-microfuge tube with a hole drilled on the cap. A yellow-tip pipette was then cut to fit in the hole so that the tops were flush against the cap of the microfuge and cut on the bottom so that flies would be able to enter. For the odorants, we cut pieces of Fisherbrand medium porosity filter paper (Hampton, NH, Catalog No. 09-801E) into 2.5 × 2 cm squares. We used Sigma-Aldrich 95% Farnesol (St. Louis, MO) diluted in Paraffin Oil. For the low concentration Farnesol we used 0.1% and for the high concentration of Farnesol we used 10%. For ethanol, we used a 5% (w/v) solution in water. We pipetted 35 μL of each odorant onto the filter paper squares and folded them up into the microfuge tubes. The paper was pushed to the bottom of the tube to prevent obstruction of the yellow tip pipettes. For the tubes with single odorants, we added the solvent in the opposite tube (e.g. Farnesol in Paraffin oil + Ethanol in water vs. Ethanol in water and Paraffin oil). For the no odorant tubes we used both paraffin oil and water. For each test, we used 20 male Canton S flies that were 4–5 days old. We aspirated the flies into the testing arenas and left them for about 12 hr overnight then placed the whole arena in −20°C kill the flies prior to counting the number of flies in each trap.

### Ethanol assay kit

We used a Megazyme Ethanol Assay Kit Cat # K-ETOH (Megazyme, Bray, CO) to measure BACs (limit of detection 0.093 mg/L). About 40 flies were treated with ethanol or air, then homogenized in

ddH$_2$O and centrifuged at 10,000 xg for 10 min. The supernatant was taken and used to measure ethanol concentrations. A negative control without fly homogenate was also used. For concentration calculations, all flies were estimated to contain 1 µL of water (calculated from previous data in *Park et al., 2018*).

## Single sensillum recording electrophysiology

Single Sensillum extracellular electrophysiology was conducted according to *de Bruyne et al., 1999* using 3–5 days old w[1118] flies. Flies were assayed under a constant stream of charcoal filtered air (36 ml/min, 22–25˚C) to prevent any contamination from environmental odors. cVa was diluted in paraffin oil (1% dilution); 35 µl was applied to filter paper and inserted into a Pasteur pipette; air was passed over the filter and presented as the stimulus. We used 1% cVa because the responses it evokes is most functionally relevant, as the magnitude of response it evokes is similar to exposing a virgin female to a male fly. Signals were amplified 1000x, fed into a computer via a 16-bit analog-to-digital converter (USB-IDAC system; Syntech), and analyzed offline with AUTOSPIKE software. The low cut-off filter setting was 200 Hz and the high cut-off setting was 3 kHz. Action potentials were recorded by inserting a glass electrode in the base of a sensillum. Data analysis was performed as reported by *Xu et al., 2005*. Signals were recorded starting 10 s before odorant stimulation. cVa-evoked action potentials were counted by subtracting the number of spikes 1 s before cVa stimulation from the spike number 1 s after cVa stimulation (Spikes/sec). The recordings were performed from separate sensilla with a maximum of two sensilla recorded from any single fly.

Ethanol was delivered by adding it into the conical flask that feeds into the IDAC. The ethanol was made from 200 proof ethanol and diluted in ddH$_2$O to make either 30% or 5% ethanol (w/v) and always covered with parafilm. For acute treatments, the flow rate was roughly 36 ml/min, 22–25˚C at roughly 2.5 L /min.

Spontaneous Activity = Total number of spikes 10 s prior to cVa delivery/10 s and ΔSpikes = Evoked activity (1 s during and after cVa delivery) – Spontaneous Activity.

## Statistical methods

For data with multiple comparisons we used a One-way ANOVA with Dunnett's correction for multiple comparisons. To test for normality of the data we used a Shapiro-Wilk's test. If one of the datasets contained non-normally distributed data we used a Kruskal-Wallis test with Dunn's correction.

# Additional information

### Funding

| Funder | Grant reference number | Author |
|---|---|---|
| National Institute on Alcohol Abuse and Alcoholism | 2R01AA01803706A1 | Nigel S Atkinson |
| National Institute on Alcohol Abuse and Alcoholism | F31AA027160 | Annie Park |
| National Institute on Alcohol Abuse and Alcoholism | T32AA07471 | Annie Park |
| National Institutes of Health | R01DC015230 | Dean P Smith |
| National Institutes of Health | 5T32GM008203 | Elizabeth A Scheuermann |

The funders had no role in study design, data collection and interpretation, or the decision to submit the work for publication.

### Author contributions

Annie Park, Conceptualization, Resources, Data curation, Software, Formal analysis, Supervision, Funding acquisition, Validation, Investigation, Visualization, Methodology, Writing - original draft, Project administration, Writing - review and editing; Tracy Tran, Data curation, Formal analysis, Validation, Writing - review and editing; Elizabeth A Scheuermann, Data curation, Formal analysis, Validation; Dean P Smith, Resources, Formal analysis, Supervision, Funding acquisition, Writing - review

and editing; Nigel S Atkinson, Conceptualization, Resources, Software, Formal analysis, Supervision, Funding acquisition, Validation, Investigation, Writing - original draft, Project administration, Writing - review and editing

## Author ORCIDs
Annie Park https://orcid.org/0000-0001-5618-2286
Dean P Smith https://orcid.org/0000-0002-4271-0436
Nigel S Atkinson https://orcid.org/0000-0001-8963-7478

## Decision letter and Author response
Decision letter https://doi.org/10.7554/eLife.59853.sa1
Author response https://doi.org/10.7554/eLife.59853.sa2

## Additional files

### Supplementary files
• Source data 1. Source Data for Figures.
• Transparent reporting form

### Data availability
All data generated or analysed during this study are included in the manuscript and supporting files.

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
