## [Decision Letter]

**Acceptance summary:**

Park et al. demonstrate a relationship between alcohol and aggression that may have some relevance both to the ecology of flies, and, perhaps, humans. It is a creative careful study that mixes detailed behavioral assessment with electrophysiological findings to explore a novel topic. This work is an important launching point for what we expect to be many future studies that build on this initial observation.

**Decision letter after peer review:**

Thank you for submitting your article "Alcohol potentiates a pheromone signal in flies" for consideration by *eLife*. Your article has been reviewed by three peer reviewers, and the evaluation has been overseen by EIC Michael Eisen acting as the Senior and Reviewing Editor. The following individuals involved in review of your submission have agreed to reveal their identity: Karla R Kaun (Reviewer #1), Fred W Wolf (Reviewer #2).

The reviewers have discussed the reviews with one another and the Reviewing Editor has drafted this decision to help you prepare a revised submission.

Park et al. provide a short report describing the phenomenon that exposure to alcohol present in food in a small chamber with decapitated virgins increases male-male aggressive behaviors. They also found that wafting alcohol vapor over the antenna of a fly prior to exposure to cVa increases activity of T1 cVa sensing neurons. Alcohol vapor also increased activity of ai2 farnesol-sensitive sensilla, and when presented in combination with farnesol in a two-choice odor test, flies preferred the alcohol+farnesol combination over farnesol or alcohol alone. The authors conclude that alcohol potentiates cVa signaling, and allude to this being the cause of increased aggression. They also conclude that alcohol increases attractiveness of a food-odor which would be relevant to flies because a food substrate may where flies can find mates. The results are interesting and of potential ecological relevance. The data is tight and presented in a transparent matter which provides confidence in the rigor of the study.

However, as written, the paper is focused on the ecological consequences of low dose ethanol on aggression, and while the data are very interesting and are an important contribution to the literature, the authors make conclusions about the ecological significance of their study that are not fully established here. With minor revisions to the text, we believe the study with the current data as is, will provide a strong addition to our understanding of how ecological concentrations of ethanol affect the physiology of olfactory response and aggressive behavior.

We recommend the following changes:

1) Reframing the manuscript so that speculation about the role of alcohol on aggression in the wild emerges in the Discussion.

2) Incorporating a more comprehensive perspective on how ethanol, or other food odors, can alter behavior and preference in *Drosophila*.

3) Add information from the literature of the role of aggression in the ecological niche (ie fermenting fruit).

4) Avoid conflating low and high dose ethanol experiments as they may induce different physiological and thus behavioral responses.

Comments and questions that emerged in review:

1) The data support the conclusions of the study. Although a direct connection between cVA and aggression wasn't presented in the current study, it has been previously published, so it makes sense the authors would conclude that alcohol increases aggression via altering cVA sensitivity. The data showing it also enhances sensitivity to food odors provides an important ecological context and suggests that alcohol may play a greater role in altering sensitivity to odors in general.

A question that arises from this, is whether this response is unique to alcohol or if it is the result of a combination or series of odors presented. Would a similar effect arise if the flies were presented with a different appetitive food odor (ethyl acetate for example) instead of alcohol? If the authors found similar results, then the effect they characterize here might be a much more general mechanism through which flies find and complete for appetitive resources (food, mates, etc). Although I think that this would be a relevant experiment to add and would strengthen the merit and general interest of the paper, the authors should not be obligated to perform this experiment since the paper stands on its own merit as an important addition to the literature. However, if the authors chose not to add this experiment, they may want to consider this possibility in the Discussion.

2) There is a relatively broad literature of the effects of natural food odors on behavioral choice in *Drosophila*, but the authors don't incorporate this into their paper. I recommend incorporating a more comprehensive perspective on how ethanol, or other food odors, can alter behavior and preference in *Drosophila*. If the authors agree, I think this would strengthen their argument substantially. Note that since some of these papers are older and some in ecologically-oriented journals, they are likely not in PubMed.

Similarly, is anything known about aggression in the ecological niche (fermenting fruit)? Zhu, Park and Baker, 2003 does not address this. Can cVA be detected in the occupied niche? How much does this affect the interpretation of the results.

Finally, in the realm of context, how do the results tie in with what is known about the broader literature on alcohol's neuronal effects. For instance, are these findings consistent with alcohol's known stimulant and depressant activities? e.g. Smoothy, R., Berry, M.S. Time course of the locomotor stimulant and depressant effects of a single low dose of ethanol in mice. Psychopharmacology 85, 57-61 (1985). https://doi.org/10.1007/BF00427322 Presumably the field has come a long way since 1985. What is currently known? Please clarify the novel insights from the present study.

3) The manuscript presupposes that LUSH is the connection between alcohol and aggression: why not test LUSH mutants? Does low concentration alcohol still promote aggression when the flies are unable to smell (a simple and classic experiment is to surgically remove the antennae)?

4) Many experiments are done at alcohol concentrations well above what flies encounter in nature. Outside of Figure 1, with interesting effects of low alcohol concentration on fighting, there are two data points in 2D that drive an effect at the low concentration. It's more like there is one paper in Figure 1, and a different one in Figures 2 and 3. It's not clear that there is any connection between the parts of the paper. This disconnect should be addressed in a revised manuscript.

5) Please address the following questions about In Figure 2: Why are cVA and alcohol presented in serial? In the wild, they would be encountered simultaneously? Why do the authors use an even higher concentration of alcohol instead of a longer presentation of 5% alcohol? Could you comment on why it appears that alcohol is increasing baseline activity of the recorded neurons instead of spike properties?

6) Figure 3 begs the question about the generalizability of alcohol exposure on all olfactory tuning. If it is general, then the arguments for niche specific amplification of food and fighting signals falls apart.

---

## [Author Response]

Park et al. provide a short report describing the phenomenon that exposure to alcohol present in food in a small chamber with decapitated virgins increases male-male aggressive behaviors. They also found that wafting alcohol vapor over the antenna of a fly prior to exposure to cVA increases activity of T1 cVA sensing neurons. Alcohol vapor also increased activity of ai2 farnesol-sensitive sensilla, and when presented in combination with farnesol in a two-choice odor test, flies preferred the alcohol+farnesol combination over farnesol or alcohol alone. The authors conclude that alcohol potentiates cVA signaling, and allude to this being the cause of increased aggression. They also conclude that alcohol increases attractiveness of a food-odor which would be relevant to flies because a food substrate may where flies can find mates. The results are interesting and of potential ecological relevance. The data is tight and presented in a transparent matter which provides confidence in the rigor of the study.However, as written, the paper is focused on the ecological consequences of low dose ethanol on aggression, and while the data are very interesting and are an important contribution to the literature, the authors make conclusions about the ecological significance of their study that are not fully established here. With minor revisions to the text, we believe the study with the current data as is, will provide a strong addition to our understanding of how ecological concentrations of ethanol affect the physiology of olfactory response and aggressive behavior.We recommend the following changes:1) Reframing the manuscript so that speculation about the role of alcohol on aggression in the wild emerges in the Discussion.

The reviewers indicate that stated ecological consequences and conclusions are speculative and should be restricted to the Discussion. We agree that in this paper there are no experiments explicitly testing these behaviors in the wild. However, the mention of these points in the Abstract and introductory text meant to provide the intellectual motivation for performing these studies and were not originally meant to be conclusions. We have tried to present these motivating ideas in a less muddled way. To address the concerns of the reviewers, we reworded the Abstract and the last paragraph of the paper.

2) Incorporating a more comprehensive perspective on how ethanol, or other food odors, can alter behavior and preference in *Drosophila*.

In the last paragraph of the manuscript we summarize what is known about alcohol’s systemic effects on fly behavior. We also describe the interaction between cVa, ethanol, and food to acknowledge the possibility that these effects could extend to other food odors and pheromones.

3) Add information from the literature of the role of aggression in the ecological niche (ie fermenting fruit).

Unfortunately, we are not aware of any studies that demonstrate the role of aggression in a natural ecological niche. The one study we do know of is Soto-Yéber, L., Soto-Ortiz, J., Godoy, P., and Godoy-Herrera, R. (2018). The behavior of adult *Drosophila* in the wild. *PloS one*, *13*(12), e0209917. Soto-Yéber et al. examined how flies behave on a few different niches and noted they did not observe aggression on the grape leaves or grains of grape. The researchers did not record their observations on video and based on the images there are too many flies to keep track of (Figures 2-3). However, the absence of evidence is not the evidence of absence, and the authors note that there are very few studies that examine fly behavior in a natural setting.

4) Avoid conflating low and high dose ethanol experiments as they may induce different physiological and thus behavioral responses.

We have specified low and high alcohol more carefully.

Comments and questions that emerged in review:1) The data support the conclusions of the study. Although a direct connection between cVA and aggression wasn't presented in the current study, it has been previously published, so it makes sense the authors would conclude that alcohol increases aggression via altering cVA sensitivity. The data showing it also enhances sensitivity to food odors provides an important ecological context and suggests that alcohol may play a greater role in altering sensitivity to odors in general.A question that arises from this, is whether this response is unique to alcohol or if it is the result of a combination or series of odors presented. Would a similar effect arise if the flies were presented with a different appetitive food odor (ethyl acetate for example) instead of alcohol? If the authors found similar results, then the effect they characterize here might be a much more general mechanism through which flies find and complete for appetitive resources (food, mates, etc). Although I think that this would be a relevant experiment to add and would strengthen the merit and general interest of the paper, the authors should not be obligated to perform this experiment since the paper stands on its own merit as an important addition to the literature. However, if the authors chose not to add this experiment, they may want to consider this possibility in the Discussion.

There is a possibility that alcohol acts as a general positive modulator of other odorants encountered by flies. However, if true this would not weaken the importance of this report. Flies are innately attracted to alcohol at low to moderate concentrations. When flies are on the fermenting food patch, they will encounter other odors that could be potentiated by alcohol and accelerate relevant behaviors such as feeding, fighting, and egg-laying.

One experiment we elected to omit from this paper was a courtship assay in which we measured courtship behavior (unilateral wing extensions) of male flies on ethanol laden food. Male courtship and copulation behavior are regulated by female pheromones methyl laurate (ML), methyl myristate (MM), and methyl palmitate (MP) Dweck et al., 2015. However, when we measured male courtship on ethanol food, there was no change in courtship or copulation behavior as measured by Unilateral Wing Extensions (UWEs), suggesting that 5% ethanol does not potentiate all odors and behaviors (Author response image 1).

We modified the last paragraph of the paper to account for the idea this response might generalize to other food odors.

**Author response image 1. sa2fig1:** 

2) There is a relatively broad literature of the effects of natural food odors on behavioral choice in *Drosophila*, but the authors don't incorporate this into their paper. I recommend incorporating a more comprehensive perspective on how ethanol, or other food odors, can alter behavior and preference in *Drosophila*. If the authors agree, I think this would strengthen their argument substantially. Note that since some of these papers are older and some in ecologically-oriented journals, they are likely not in PubMed.

It is difficult to review literature in such a short report format. However, we have modified the last paragraph to discuss this idea.

Similarly, is anything known about aggression in the ecological niche (fermenting fruit)? Zhu, Park and Baker, 2003 does not address this. Can cVA be detected in the occupied niche? How much does this affect the interpretation of the results.

There is not as much known about aggression in an ecological niche (See point 3 above). But there is some literature that describes the natural deposition of cVa in the environment. In Keesey et al., 2016, the researchers found that deposits made on blueberries include ML, MM, MP, cVa and other cuticular hydrocarbons using GC-MS. In another paper by Mercier et al., 2018. Researchers found that male flies will deposit cVa in a particular spot on the arena and spend more time around the deposit they made (<16 mins). We have included this information in the manuscript. However, there is no literature to our knowledge about detecting cVa deposits on fermented fruit patches specifically.

Finally, in the realm of context, how do the results tie in with what is known about the broader literature on alcohol's neuronal effects. For instance, are these findings consistent with alcohol's known stimulant and depressant activities? e.g. Smoothy, R., Berry, M.S. Time course of the locomotor stimulant and depressant effects of a single low dose of ethanol in mice. Psychopharmacology 85, 57-61 (1985). https://doi.org/10.1007/BF00427322 Presumably the field has come a long way since 1985. What is currently known? Please clarify the novel insights from the present study.

We believe the reviewers are suggesting that alcohol odor could also increase locomotor activity in flies, which could inflate the number of aggressive encounters. Stimulatory effects of low doses of alcohol have been reported in flies (Wolf et al., 2002). To address this, we added a line-crossing experiment to the paper, which provides a measure of locomotor activity (Figure 1H). We performed the line crossing assays within the aggression chambers with alcohol added into the food and found no correlation between locomotor activity and aggression. Flies exposed to 5% alcohol food did not increase their locomotor activity and they also did not have an increase in BAC, but they did show an increase in multiple aggressive behaviors. We did see a very slight decrease in locomotion in flies on 20% alcohol food. However, at this concentration aggressive behaviors are not changed. Thus, it appears that our alcohol treatments are not increasing aggression simply by acting as stimulants.

3) The manuscript presupposes that LUSH is the connection between alcohol and aggression: why not test LUSH mutants? Does low concentration alcohol still promote aggression when the flies are unable to smell (a simple and classic experiment is to surgically remove the antennae)?

We were dissuaded from doing this experiment because the cVa pathway itself is necessary for aggression (Wang et al., 2010). Although we did not test LUSH mutants, we did attempt to assay Or67d mutants and saw they were not aggressive (Author response image 2 shows lunges of Or67d receptor mutants with CantonS controls). Or67d is the receptor that hetero-dimerizes with ORCO and SNMP, which binds to ligand-bound LUSH. Although there is no increase in aggression following exposure to 5% alcohol food in Or67d -/- flies, the flies are not aggressive in the first place and so this cannot be interpreted to mean that the cVa sensing pathway is necessary for alcohol-induced aggression. It was this Or67d receptor mutant experiment that made the LUSH experiment appear uninformative.

One approach to confirm that LUSH mediates alcohol-induced aggression would be to mutate the alcohol-binding sites of LUSH (identified in Kruse et al., 2003). Ideally, the mutagenesis would only affect alcohol binding and leave cVa binding unimpaired. However, we believe this is outside the scope of this short reports paper and is not of critical importance for this study.

4) Many experiments are done at alcohol concentrations well above what flies encounter in nature. Outside of Figure 1, with interesting effects of low alcohol concentration on fighting, there are two data points in 2D that drive an effect at the low concentration. It's more like there is one paper in Figure 1, and a different one in Figures 2 and 3. It's not clear that there is any connection between the parts of the paper. This disconnect should be addressed in a revised manuscript.

Figure 1 and 2 both share a low concentration of alcohol (5%) that can occur in nature. However, as detailed in the manuscript the concentration of alcohol in the air in Figure 1 and Figures 2 and 3 is probably never identical. The reason for this is that in the behavioral assays in Figure 1 the alcohol vapor is generated by passive evaporation from the food dish (% alcohol refers to concentration in the food) and was never indended to exactly equal the alcohol in the ephys experiments in which the alcohol vapor was generated using a bubbler (in the ephys, the % alcohol refers to the concentration in the solution through which air is bubbled). While this was described in the original document, we have made a small adjustment that we think helps make this clearer. We think that this addresses the apparent disconnect between Figure 1 and Figures 2 and 3.

In Figure 2D post 5% Alc the two largest value data points do not meet the formal definition of outliers. Furthermore, even if these two data points are removed, the pre- and post-treatment statistical significance is still maintained. Therefore, they are not driving the effect (raw data in the Source data 1).

However, in re-examining these data we realized that the largest value in Figure 2D post 30% Alc is actually a statistical outlier. Thank you for helping us find this. Removing it does not eliminate the significant difference between post-30% and pre or during treatment. Therefore, we have removed this outlier. This also has the beneficial effect of making the relationship between all of the datapoints easier to see in the graph.

5) Please address the following questions about In Figure 2: Why are cVA and alcohol presented in serial? In the wild, they would be encountered simultaneously? Why do the authors use an even higher concentration of alcohol instead of a longer presentation of 5% alcohol? Could you comment on why it appears that alcohol is increasing baseline activity of the recorded neurons instead of spike properties?

Serial presentation occurred because the rig was designed for serial presentation. However, we do not think that this limitation invalidates the work because on approach to a food patch alcohol would more likely be encountered before cVa, as cVa has a relatively low range of detection (about 2.5 mm from Figure 2F in Mercier et al., 2018).

Examining the responses after a longer odor presentation might have been a good idea and might have revealed other aspects of the response. At the time, the value of a longer exposure was not apparent to us. However, it is not just the baseline that is altered, as spike duration changes as well.

Thirty percent was chosen because other work on the systemic effects of alcohol had used 30%. While the other work ended up being irrelevant to this paper we had completed a body of work using 30% ethanol. Having said this, one should keep in mind that 5% and 30% bubbler ethanol in the ephys experiments should be viewed as low and high concentrations when compared to the behavioral experiments that used 5% and 20% ethanol in the food. It is unlikely that identical concentrations of ethanol in a bubbler and in the food result in the exact same concentration in the air. While this lack of a one-to-one relationship is not what one would want in a perfect world, we still think that we have documented behavioral responses and ephys responses to low concentration alcohol that support a single unambiguous interpretation.

We do not have any data that allow us to do anything other than speculate on the origins of the shift in activity with alcohol. We might imagine that mechanistically alcohol is either (1) binding to LUSH and stabilizing it in its active confirmation causing it to activate the Or67d/Orco complex in the absence of cVa OR (2) ethanol could also be directly acting on Orco as an allosteric modulator to increase its spontaneous openings. Because the farnesol-sensing neurons do not have an identified OBP, it is difficult to say if (1) is true for all alcohol-sensitive OSNs. However, both farnesol and cVa-sensing neurons (and most OSNs) have Orco which would mean that the behavioral responses to olfactory alcohol are broadly tuned. Examination of these hypotheses require future experiments to elucidate how alcohol is potentiating these pathways.

6) Figure 3 begs the question about the generalizability of alcohol exposure on all olfactory tuning. If it is general, then the arguments for niche specific amplification of food and fighting signals falls apart.

Perhaps, alcohol does augment all odor detection. But we disagree with the premise that such a phenomenon would negate our niche-specific arguments. Even if olfaction was enhanced across the board it could still result in niche-specific changes in behavior such as increased attraction to reproductive resources and increased aggressive responses to other males. Also see 1) in which we find that courtship behavior in males on 5% ethanol food does not change.